# BMI-mediated association between glyphosate exposure and increased risk of atherosclerotic heart disease: A large-scale cross-sectional study

**Wendi Xu[1], Zhe Chen[2], Yuhong Jiang[3], Hongbo Zeng[4], Nan He[5], Ziyi Liu[6]\*, Meirong Zhou[1]\***

**1** Department of General Surgery, The Second Xiangya Hospital of Central South University, Changsha, Hunan, China, **2** Department of Thoracic Surgery, The Second Xiangya Hospital of Central South University, Changsha, Hunan, China, **3** The Department of Integrated Traditional Chinese and Western Medicine, National Clinical Research Center for Geriatrics, West China Hospital, Sichuan University, Chengdu, Sichuan, China, **4** Department of Urology, The Second Xiangya Hospital of Central South University, Changsha, Hunan, China, **5** The Third People's Hospital of Hengyang City, Hengyang, Hunan, China, **6** Department of Emergency Medicine, The Second Xiangya Hospital of Central-South University. Emergency and Difficult Diseases institute of Central South University, Changsha, Hunan, China

\* 228212239@csu.edu.cn (ZL); zmr247@csu.edu.cn (MZ)

**Data Availability Statement:** All data used for analysis are publicly available on the NHANES website, and data are available at https://wwwn.cdc.gov/nchs/nhanes/search/default.aspx.

## Abstract

### Background

Glyphosate, as the main component of glyphosate pesticides, has been shown to have toxic effects on multiple human systems. However, the association between glyphosate and atherosclerotic cardiovascular disease (ASCVD) remains unclear. This study aims to explore the effect of glyphosate exposure on ASCVD.

### Methods

This study involved 1,602 participants and employed various statistical techniques, such as multivariable logistic regression, linear fitting, mediation analysis, subgroup analysis, and sensitivity analysis, to elucidate the association between glyphosate exposure and cardiovascular diseases.

### Results

Compared with the low level of glyphosate exposure, the risk of ASCVD increased significantly with high level of glyphosate exposure (OR = 2.05, 95%CI = 1.17–3.58, p<0.05, p for trend<0.05), which showed a linear upward trend. Further analysis found that exposure to high levels of glyphosate and angina pectoris (OR = 2.84, 95%CI = 1.00–8.04, p<0.05, p for trend<0.05), coronary heart disease (OR = 3.76, 95%CI = 1.80–7.83, p<0.01, p for trend<0.05), heart attack (OR = 2.65, 95%CI = 1.35–5.23, p<0.01, p for trend = 0.09) were associated. Mediating analysis found that BMI mediated the association between glyphosate exposure and ASCVD, with an indirect effect of 0.002405(95%CI: 0.000182–0.01, p =

**Funding:** The author(s) received no specific funding for this work.

**Competing interests:** The authors have declared that no competing interests exist.

0.02) and a direct effect of 0.039336(95%CI: 0.000391–0.09), total effect of 0.041741(95% CI: 0.002112–0.10).

## Conclusion

Increased exposure to glyphosate is associated with an increased risk of ASCVD, and BMI plays a mediating role in this association. In addition, glyphosate exposure is associated with a higher risk of angina, coronary heart disease, and heart attack.

## Introduction

Atherosclerotic cardiovascular disease refers to the accumulation of fats and fibrous substances in the arterial intima, forming plaques that gradually invade the arterial lumen, ultimately leading to tissue ischemia and a series of pathological changes in the heart and blood vessels [1]. ASCVD is the leading cause of death globally. According to statistics, over 17 million people died from ASCVD in 2015, accounting for about 1/3 of total deaths [1]. There are numerous risk factors for ASCVD, including poor dietary and lifestyle habits, underlying diseases, and other factors that accelerate the occurrence and progression of ASCVD [2]. Due to the extremely high mortality rate and poor prognosis of ASCVD, preventing its onset and slowing its progression are crucial. In recent years, environmental pollution and other issues have become increasingly severe, and growing evidence suggests a correlation between exposure to environmental pollutants and the progression of ASCVD [3].

Glyphosate, known as N-phosphonomethyl glycine, is the primary active ingredient in glyphosate-based herbicides [4]. Due to its broad-spectrum activity and effective suppression of weeds, these herbicides are currently used in 140 countries and regions, making them the most widely used herbicides globally [5]. The extensive use of glyphosate-based herbicides has led to widespread exposure to glyphosate, which can be detected in air, food, soil, and water [6]. Glyphosate in the environment can enter the human body through various pathways, such as skin contact, inhalation, and ingestion [7]. Previously, it was believed that glyphosate posed minimal risk to human health despite environmental exposure, because its herbicidal action mainly inhibits the shikimate pathway in plants, which is not present in vertebrates [5]. However, as research advances, increasing evidence suggests that glyphosate also has adverse effects on vertebrates. A study on guinea pigs found that glyphosate impaired their growth and reproductive function [8]. Another study on rainbow trout found that long-term exposure to low concentrations of glyphosate in the environment affected the development and metabolism of their offspring [9]. Furthermore, several large-scale cross-sectional studies conducted in recent years have shown that glyphosate is significantly associated with adverse events such as diabetes, depression, and liver dysfunction [10–12].

However, there is currently limited research on the association between glyphosate and ASCVD. Upon reviewing the literature, we only found reports linking glyphosate exposure to cardiovascular disease (CVD), and these reports yielded negative results [13]. Similarly, existing studies on the association between organophosphate (OP) exposure and cardiovascular disease present contradictory findings. A study conducted in Myanmar revealed a significantly increased risk of cardiovascular disease among workers with long-term exposure to OP pesticides compared to the non-exposed group [14]. Conversely, results from a cross-sectional study in the United States suggested no statistically significant association between

organophosphate exposure and cardiovascular disease [15]. Therefore, our team has designed this study with the aim of investigating the association between glyphosate and ASCVD.

## Methods

### Study populations

The National Health and Nutrition Examination Survey (NHANES) is a comprehensive survey of various racial groups and health-related issues in the United States. The data are obtained through a complex, stratified, multistage probability sampling design. In this study, we utilized data from participants who took part in NHANES during the period from 2013 to 2014. This research received approval from the Institutional Review Board of the National Center for Health Statistics, and all participants provided written informed consent. The study was conducted in accordance with the Declaration of Helsinki, and study protocol was approved by the NCHS Institutional Review Board (Continuation of Protocol #2011–17).

A total of 10,175 participants' data were collected for this study. Subsequently, individuals under the age of 20 (n = 4,406), pregnant women (n = 65), participants without information on heart failure (n = 5), angina (n = 8), coronary artery disease (n = 13), heart attack (n = 1), stroke (n = 4), and those without information on glyphosate exposure (n = 4,071) were excluded. Ultimately, data from 1,602 participants were analyzed in this study. Please refer to Fig 1 for the specific workflow.

### Diagnostic criteria for CVD and ASCVD

Participants' cardiovascular disease status was determined based on self-reported physician-diagnosed information obtained from the "Medical Conditions" questionnaire. This questionnaire comprised five separate questions regarding congestive heart failure, angina, coronary artery disease, heart attack, and stroke, respectively. When participants were asked whether a doctor or other health professional had ever informed them of having congestive heart failure, angina, coronary artery disease, heart attack, stroke, a positive cardiovascular disease status was assigned if the participant answered "yes" to any one or more of these questions; otherwise, it was considered negative.

According to the 2013 guidelines from the American College of Cardiology (ACC) and the American Heart Association (AHA), ASCVD is defined as a diagnosis of angina, coronary artery disease, heart attack, or stroke [16]. Therefore, we defined a positive ASCVD status as follows: when participants were asked whether a doctor or other health professional had ever informed them of having angina, coronary artery disease, heart attack, or stroke, a positive ASCVD status was assigned if the participant answered 'yes' to any one or more of these questions; otherwise, it was considered negative.

### Measurement of urinary glyphosate levels

Following collection, urine samples were transported and stored in accordance with clinical laboratory standards. In the laboratory, a combination of online 2D ion chromatography and tandem mass spectrometry techniques was employed, along with isotope dilution for quantitative measurement of urinary glyphosate levels, using 200μl of urine per assay. For results below the limit of detection of 0.2ng/ml, an interpolated value was provided, calculated as the square root of the limit of detection divided by 2 [17]. Finally, urine glyphosate concentrations were normalized to urinary creatinine levels to account for urine dilution, and were thus expressed as μg/g of creatinine [18].

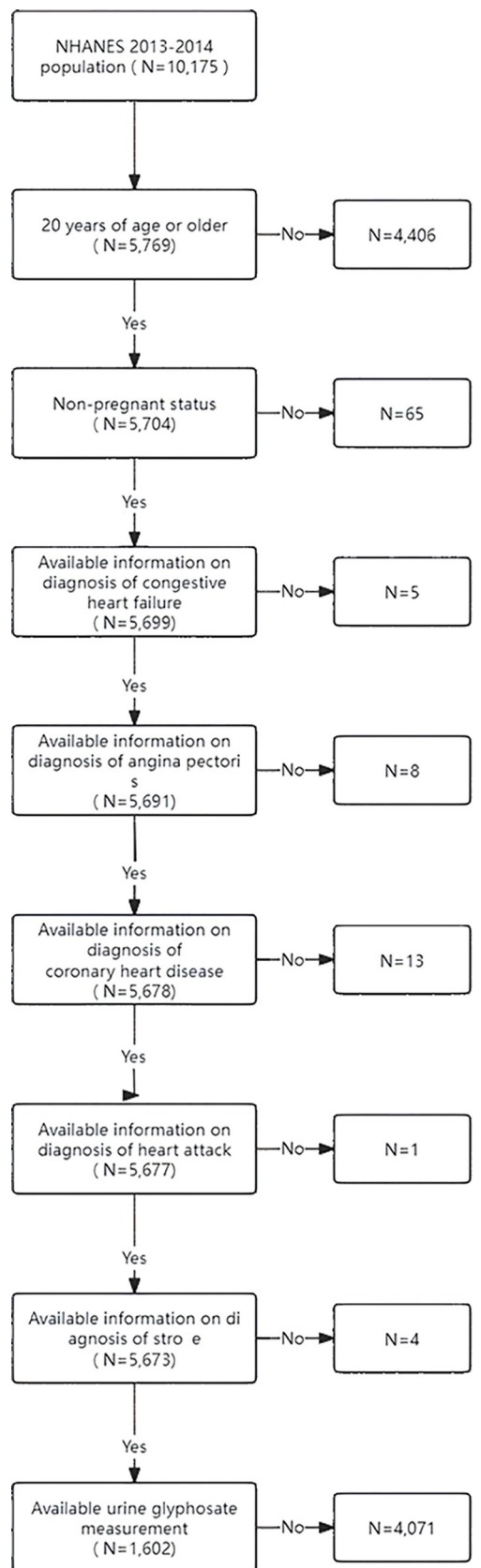

**Fig 1. Schematic flow diagram of inclusion and exclusion criteria for our study cohort.**

## Study variables

This study analyzed the following factors: age, gender, ethnicity, marital status, education, poverty income ratio (PIR), smoking (never smoked, former smoker, current smoker), alcohol consumption (never drank, past drinking history, light to moderate drinking, heavy drinking. Heavy drinking: ≥3 drinks/day for women or ≥4 drinks/day for men or binge drinking ≥5 days/month; moderate drinking: ≥2 drinks/day for women or ≥3 drinks/day for men or binge drinking ≥2 days/month; light drinking: excluding heavy and moderate drinking), activity level (engaging in moderate to vigorous physical activity, exercise, or recreational activities for more than 10 minutes per week considered active, otherwise inactive), body mass index (BMI) (kg/m$^2$), hyperlipidemia, hypertension, and diabetes.

## Statistical analysis

NHANES employs stratified multi-stage probability sampling to minimize biases introduced by stratification, non-response, and oversampling. Categorical variables are presented in baseline tables as absolute values (n) and percentages (%). To address skewness, urinary glyphosate levels were logarithmically transformed before analysis. Glyphosate concentrations were categorized into quartiles based on the weighted distribution of the sample. To investigate the potential association between environmental glyphosate exposure and ASCVD, a multivariable logistic regression analysis model was used to explore whether urinary glyphosate levels had a statistically significant relationship with CVD and ASCVD. To mitigate biases from confounding exposures, three different models were employed. Model 1 made no adjustments for any variables. Model 2 adjusted for variables such as age, gender, and race. Model 3 adjusted for all considered variables, including age, gender, race, marital status, education, poverty level, smoking, drinking, activity level, BMI, hypertension, and diabetes. After constructing the logistic regression models, cubic spline curves were plotted for results with statistical significance to display linear relationships. To delve deeper into the association between glyphosate and cardiovascular diseases, the aforementioned methods were used to analyze the relationship between urinary glyphosate levels and congestive heart failure, angina, coronary artery disease, heart attacks, and strokes individually.

To explore the potential mediating effects between glyphosate exposure and cardiovascular diseases, a multivariable linear regression model was used to conduct causal mediation analysis between glyphosate exposure and inflammatory factors, as well as BMI. Additionally, subgroup analyses were performed, stratifying by different confounding factors to assess the consistency of study results across different subgroups. Finally, several sensitivity analyses were conducted.

All analyses in this study were performed using R 4.21. It was statistically significant when the P value was set below 0.05.

## Result

### Study population

This study ultimately included 1,602 eligible participants in the final statistical analysis. To investigate the associations between glyphosate and both CVD and ASCVD separately, S1 and S2 Tables were established based on the presence of CVD and ASCVD, respectively. As shown in the tables, compared to the non-CVD and non-ASCVD groups, participants in the CVD and ASCVD groups had a higher proportion of high-level glyphosate exposure. Additionally, the proportion of participants aged over 60, Caucasian, with high BMI, smokers, and those with lower activity levels was higher. Moreover, the proportion of participants with underlying

**Table 1. The logistic regression analysis results of the association between glyphosate and CVD, ASCVD.**

| | | Model 1 | | | Model 2 | | | Model 3 | | |
|---|---|---|---|---|---|---|---|---|---|---|
| | | OR (95%CI) | P Value | P for Trend | OR (95%CI) | P Value | P for Trend | OR (95%CI) | P Value | P for Trend |
| CVD | Q1 | Ref | Ref | <0.05 | Ref | Ref | 0.10 | Ref | Ref | <0.05 |
| | Q2 | 1.30(0.79, 2.13) | 0.29 | | 1.07(0.66, 1.74) | 0.77 | | 1.18(0.73, 1.92) | 0.47 | |
| | Q3 | 1.65(0.96, 2.86) | 0.07 | | 1.42(0.87, 2.31) | 0.15 | | 1.77(0.97, 3.22) | 0.06 | |
| | Q4 | 2.46(1.36, 4.62) | <0.01 | | 1.66(0.79, 3.49) | 0.17 | | 1.89(0.96, 3.75) | 0.06 | |
| ASCVD | Q1 | Ref | Ref | <0.001 | Ref | Ref | <0.05 | Ref | Ref | <0.05 |
| | Q2 | 1.48(0.91, 2.39) | 0.11 | | 1.25(0.77, 2.02) | 0.34 | | 1.37(0.76, 2.45) | 0.27 | |
| | Q3 | 1.97(1.04, 3.74) | 0.04 | | 1.72(0.95, 3.11) | 0.07 | | 2.11(0.99, 4.52) | 0.05 | |
| | Q4 | 2.72(1.73, 4.28) | <0.001 | | 1.82(1.05, 3.13) | 0.03 | | 2.05(1.17, 3.58) | <0.05 | |

*Q1, 0–25%; Q2, 25%-50%; Q3, 50%-75%; Q4, 75–100%.

conditions such as hyperlipidemia, hypertension, and diabetes was also higher. Glyphosate exposure levels were categorized as follows: Q1: -0.8508 to -0.6021; Q2: -0.6021 to -0.4174; Q3: -0.4174 to -0.1938; Q4: -0.1938 to 0.8102. For more detailed participant information, please refer to S1 and S2 Tables.

## Association between glyphosate and CVD, ASCVD

Table 1 presents the results of the logistic regression analysis between glyphosate exposure and the prevalence of CVD and ASCVD. In the constructed regression models, Q1 served as the reference value. As shown in Table 1, there was no significant statistical correlation between urinary glyphosate concentration and the prevalence of CVD. In Model 3, after adjusting for all variables, the odds ratio (OR) and p for Q4 were (OR = 1.89, 95%CI = 0.96–3.75, p = 0.06, p for trend<0.05). Conversely, for ASCVD, there was a statistically significant correlation with urinary glyphosate concentration. Glyphosate exposure was identified as a risk factor for ASCVD, and the trend of increasing ASCVD risk with higher glyphosate exposure levels was also statistically significant. In Model 3, the OR and p for Q4 were (OR = 2.05, 95%CI = 1.17–3.58, p<0.05, p for trend<0.05).

In the logistic regression model of glyphosate and ASCVD prevalence, we observed statistically significant results. Further, we plotted a restricted cubic spline curve to fit the data. As depicted in Fig 2, in the linear fit between glyphosate and ASCVD, we found that the odds ratio (OR) for ASCVD increased initially with rising urinary glyphosate concentration levels before plateauing, showing a non-linear trend (Nonlinear P: 0.2824).

## Association between glyphosate and congestive heart failure/coronary heart disease/angina/heart attack/stroke

Following the logistic regression analysis of the association between glyphosate exposure and CVD, ASCVD, further analysis was conducted on the association between glyphosate exposure and five sub-diseases: congestive heart failure, angina, coronary artery disease, heart attack, and stroke. The results are presented in Table 2. From the table, it is evident that there is a statistically significant association between glyphosate exposure and angina, coronary artery disease, and heart attack, where glyphosate acts as a risk factor. Compared to Q1, the Model 3 Q4 for the aforementioned three diseases are: angina (OR = 2.84, 95%CI = 1.00–8.04, p<0.05, p for trend<0.05), coronary artery disease (OR = 3.76, 95%CI = 1.80–7.83, p<0.01, p for trend<0.05), heart attack (OR = 2.65, 95%CI = 1.35–5.23, p<0.01, p for trend = 0.09).

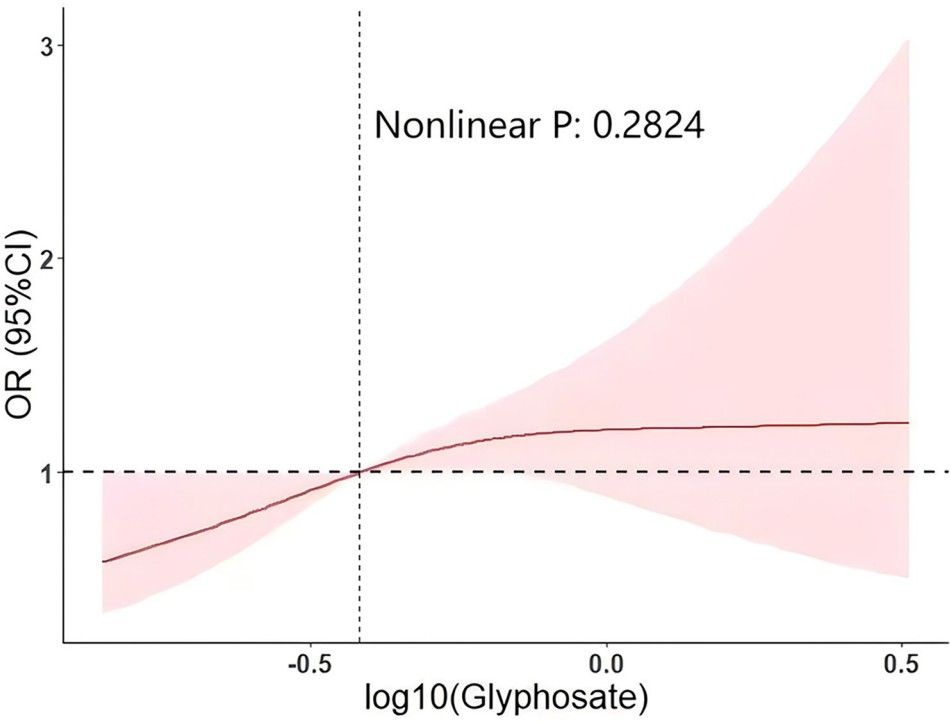

**Fig 2. The weighted restricted cubic spline curve of urinary glyphosate concentration and ASCVD prevalence.**

Additionally, no statistically significant association was found between glyphosate exposure and congestive heart failure or stroke.

## Mediation analysis

A multiple linear regression model was employed to investigate the relationship between glyphosate exposure and inflammatory markers, as well as BMI. BMI was adjusted as a covariate in the logistic regression model, while it remained unadjusted in the multiple linear regression model. The results are presented in S3 Table. Glyphosate exposure showed a positive correlation with alkaline phosphatase ($\beta$ = 3.3299, 95% CI: 0.0909–6.5689, P < 0.05) and BMI ($\beta$ = 0.7193, 95% CI: 0.1682–1.2703, P < 0.05), while no significant statistical associations were found with white blood cells, lymphocytes, or neutrophils.

Given the close association observed in the aforementioned regression models between glyphosate exposure and increased ASCVD risk, further mediation analysis was conducted to explore potential mechanisms. We evaluated the mediating effects of inflammatory markers and BMI, with results presented in S4 Table. After analyzing five potential mediating factors, we found that only BMI exhibited a statistically significant mediating effect between glyphosate exposure and increased ASCVD prevalence, with an indirect effect of 0.002405 (95% CI: 0.000182–0.01, p = 0.02), a direct effect of 0.039336 (95% CI: 0.000391–0.09), and a total effect of 0.041741 (95% CI: 0.002112, 0.10). Visual representation of these results is depicted in Fig 3.

## Analysis of subgroup

We conducted subgroup analysis to identify confounding factors that may influence the relationship between glyphosate exposure and ASCVD incidence, ensuring the stability of the association across different subgroups. A forest plot (Fig 4) was generated to illustrate our

**Table 2. The logistic regression analysis results of the association between glyphosate and congestive heart failure, angina, coronary heart disease, heart attack, stroke.**

| | | Model 1 | | | Model 2 | | | Model 3 | | |
|---|---|---|---|---|---|---|---|---|---|---|
| | | OR (95%CI) | P Value | P for trend | OR (95%CI) | P Value | P for trend | OR (95%CI) | P Value | P for trend |
| Congestive Heart Failure | Q1 | Ref | Ref | 0.40 | Ref | Ref | 0.92 | Ref | Ref | 0.97 |
| | Q2 | 0.81(0.28, 2.30) | 0.67 | | 0.62(0.20, 1.88) | 0.37 | | 0.64(0.21, 1.93) | 0.41 | |
| | Q3 | 0.72(0.22, 2.33) | 0.56 | | 0.54(0.18, 1.65) | 0.26 | | 0.58(0.21, 1.61) | 0.27 | |
| | Q4 | 1.65(0.53, 5.11) | 0.36 | | 1.01(0.25, 4.13) | 0.99 | | 0.98(0.28, 3.45) | 0.98 | |
| Coronary Heart Disease | Q1 | Ref | Ref | <0.05 | Ref | Ref | 0.1 | Ref | Ref | <0.05 |
| | Q2 | 0.43(0.13, 1.37) | 0.14 | | 0.33(0.11, 1.00) | 0.05 | | 0.35(0.10, 1.18) | 0.86 | |
| | Q3 | 0.53(0.23, 1.23) | 0.13 | | 0.42(0.19, 0.93) | <0.05 | | 0.54(0.27, 1.10) | 0.85 | |
| | Q4 | 3.34(1.17, 9.51) | <0.05 | | 2.16(0.68, 6.83) | 0.17 | | 2.84(1.00, 8.04) | <0.05 | |
| Angina | Q1 | Ref | Ref | <0.001 | Ref | Ref | <0.05 | Ref | Ref | <0.05 |
| | Q2 | 3.61(1.90, 6.86) | <0.001 | | 3.16(1.57, 6.35) | <0.01 | | 3.65(1.95, 6.82) | <0.001 | |
| | Q3 | 2.90(1.26, 6.71) | <0.05 | | 2.36(0.97, 5.75) | 0.06 | | 2.87(0.88, 9.40) | 0.08 | |
| | Q4 | 5.67(2.64, 12.16) | <0.001 | | 3.57(1.64, 7.76) | <0.01 | | 3.76(1.80, 7.83) | <0.01 | |
| Heart Attack | Q1 | Ref | Ref | <0.05 | Ref | Ref | 0.11 | Ref | Ref | 0.09 |
| | Q2 | 2.24(0.96, 5.23) | 0.06 | | 1.96(0.78, 4.93) | 0.14 | | 2.20(0.82, 5.91) | 0.11 | |
| | Q3 | 2.41(0.99, 5.87) | 0.05 | | 2.07(0.91, 4.68) | 0.08 | | 2.53(1.05, 6.13) | <0.05 | |
| | Q4 | 3.74(1.92, 7.28) | <0.001 | | 2.41(1.31, 4.46) | <0.01 | | 2.65(1.35, 5.23) | <0.01 | |
| Stroke | Q1 | Ref | Ref | 0.36 | Ref | Ref | 0.93 | Ref | Ref | 0.62 |
| | Q2 | 0.67(0.20, 2.24) | 0.49 | | 1.25(0.16, 1.72) | 0.27 | | 0.56(0.17, 1.83) | 0.31 | |
| | Q3 | 1.58(0.41, 6.06) | 0.48 | | 1.30(0.39, 4.36) | 0.65 | | 1.96(0.73, 5.22) | 0.16 | |
| | Q4 | 1.18(0.49, 2.87) | 0.69 | | 0.71(0.30, 1.73) | 0.43 | | 0.89(0.31, 2.60) | 0.82 | |

*Q1, 0–25%; Q2, 25%-50%; Q3, 50%-75%; Q4, 75–100%.

findings. From the plot, it is evident that the interactions between smoking (P = 0.048), physical activity (P<0.001), hypertension (P = 0.002), and glyphosate exposure significantly affect the risk of ASCVD incidence. Furthermore, the horizontal lines for non-smokers (OR: 1.924, 95% CI: 1.325–2.794), non-hypertensive individuals (OR: 2.399, 95% CI: 1.568–3.669), and physically active participants (OR: 2.399, 95% CI: 1.568–3.669) do not intersect with the null vertical line and are positioned to the right of it. The interactions with other confounding factors were found to be non-significant (P<0.05).

## Analysis of sensitivity

Finally, we conducted three sensitivity analyses to demonstrate the robustness of the results: 1) adjusting for three liver function indicators: gamma-glutamyl transferase, alanine aminotransferase, and aspartate aminotransferase; 2) adjusting for several kidney function indicators: blood creatinine, blood urea nitrogen, glomerular filtration rate, albumin-to-creatinine ratio, and uric acid; 3) removing missing values. After adjusting for these variables, the results remained unchanged. Please refer to S5 Table for details.

## Discussion

In this cross-sectional study, we utilized the NHANES database to investigate the association between glyphosate exposure and CVD, yielding highly meaningful results. By constructing regression models, we identified glyphosate exposure as a risk factor for ASCVD, and linear fitting revealed an increasing trend in ASCVD incidence with higher levels of glyphosate

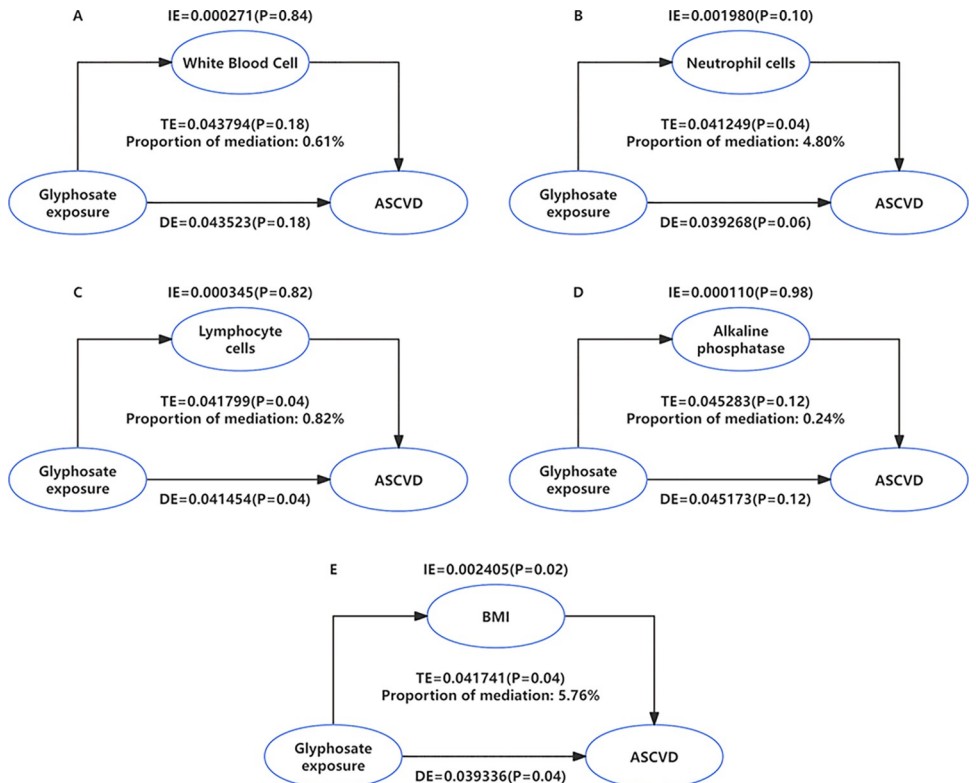

**Fig 3. Visualization of the mediating effects of inflammatory factors and BMI on the association between glyphosate exposure and ASCVD prevalence.** A. Mediation models of Glyphosate, ASCVD and white blood cell; B. Mediation models of Glyphosate, ASCVD and neutrophil cells; C. Mediation models of Glyphosate, ASCVD and lymphocyte cells; D. Mediation models of Glyphosate, ASCVD and alkaline phosphatase; E. Mediation models of Glyphosate, ASCVD and BMI.

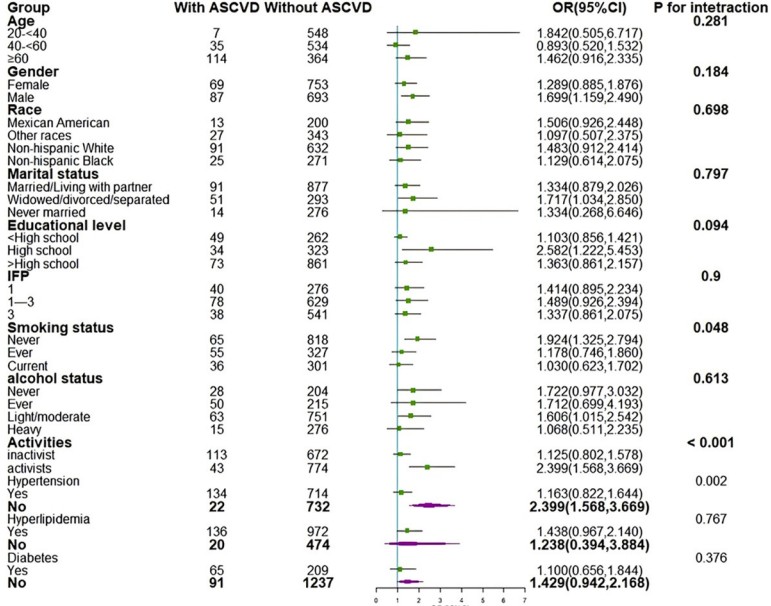

**Fig 4. Relationship between glyphosate exposure and ASCVD risk in different subgroups.**

exposure. Further analysis of five diseases—heart failure, coronary heart disease, angina, heart attack, and stroke—indicated statistically significant associations between glyphosate exposure and coronary heart disease, angina, and heart attacks. In the mediation analysis, we discovered that BMI mediated the association between glyphosate exposure and increased risk of ASCVD incidence.

Research on the link between glyphosate exposure and cardiovascular diseases has investigated the relationship between glyphosate exposure and both overall cardiovascular disease rates and specific diseases. The results suggest that glyphosate exposure increases the risk of atherosclerotic cardiovascular diseases. Moreover, there is evidence linking glyphosate exposure to higher rates of angina pectoris, coronary artery, and heart attack.

In our study, we initially used regression models to analyze the association between glyphosate exposure and CVD, but we did not find any statistically significant link between the two. Building upon this, we excluded congestive heart failure as a target disease and specifically examined the association between glyphosate exposure and ASCVD, which revealed glyphosate exposure as a risk factor for ASCVD. We hypothesized that this association might be due to glyphosate's role in promoting the formation and exacerbation of atherosclerosis. Reviewing online journals, we found numerous studies indicating that glyphosate exposure accelerates the production of reactive oxygen species (ROS), which induce oxidative damage to DNA, lipids, and proteins [19,20], promote the oxidation of low-density lipoprotein (LDL) to oxidized LDL, thus facilitating atherosclerosis formation [1]. Additionally, an animal study on zebrafish suggested that the antioxidant compound Uncaria tomentosa has a protective effect against glyphosate-induced oxidative stress [21]. Furthermore, glyphosate can promote ASCVD by inducing cell apoptosis. Apoptosis of vascular smooth muscle cells promotes calcification and internal deformation of arterial plaques, preventing vascular remodeling but exacerbating the narrowing of arteries [22]. Moreover, research by Young-hee Kim et al. investigated the toxicity of glyphosate in combination with polyoxyethyleneamine (TN-20), a common surfactant in glyphosate herbicides. They found that this combination enhances glyphosate toxicity by inducing cell apoptosis and necrosis through mitochondrial damage, accelerating cell death [23]. Therefore, we consider glyphosate-induced apoptosis of vascular smooth muscle cells as another significant factor contributing to ASCVD. It's worth noting that animal experiments suggest glyphosate exposure may lead to enlargement of the zebrafish ventricle, which could disrupt normal heart function and potentially cause circulatory blockages [24].

Heart failure typically refers to the heart's inability to pump enough blood to meet the body's needs, resulting in a range of symptoms related to congestion and ischemia, such as shortness of breath, leg swelling, fatigue, and poor peripheral circulation [25]. Based on current experimental findings and previous literature, we are inclined to believe that glyphosate exposure accelerates atherosclerosis, leading to a series of cardiovascular diseases, meaning that the impact of glyphosate exposure on heart failure is relatively minor. Although atherosclerotic cardiovascular disease can also lead to heart failure, atherosclerosis plays an indirect rather than a direct role in this process [25]. We believe this is the reason for the statistically negative results. Upon reviewing relevant literature, we only found one report on the association between glyphosate exposure and CVD, which also indicated no statistically significant association between glyphosate exposure and CVD risk. Unfortunately, this study did not further investigate the association between glyphosate and ASCVD [13]. Additionally, this paper uses the same definition of CVD as our study. Additionally, other research suggests that glyphosate can cause arrhythmias, leading to QTc prolongation, arrhythmias, and atrioventricular block [26].

Our research also yielded negative results regarding the relationship between glyphosate exposure and stroke. Stroke refers to acute focal neurological deficits caused by vascular factors, resulting in impaired neurological function such as visual and motor impairments [27].

There are many causes of stroke, including atherosclerosis, deep intracerebral hemorrhage, post-infectious inflammatory arteries, intracranial masses, and other diseases that can lead to adverse outcomes [27]. Atherosclerotic lesions are just one of the causes of stroke, which may explain the lack of meaningful association between glyphosate and stroke. Upon reviewing literature on glyphosate and brain diseases, we only found one case report of unilateral hippocampal infarction following glyphosate ingestion [28], as well as reports of glyphosate affecting the gut-brain axis in animals, impacting animal social behavior [29], and accelerating the onset and progression of neurodegenerative diseases [24,30]. No other studies linking glyphosate to stroke were found.

We believe that glyphosate increases the risk of ASCVD by promoting atherosclerosis, a view indirectly supported by our mediation analysis. The results of the mediation analysis suggest that BMI mediates the association between glyphosate exposure and ASCVD. However, we have concerns about the other results of the mediation analysis. Theoretically, inflammatory reactions can alter the function of arterial wall cells, promoting the occurrence and progression of atherosclerosis [1], while glyphosate exposure can induce inflammation and oxidative stress [31]. Surprisingly, our analysis of the mediation effects of four inflammatory factors—white blood cells, neutrophils, lymphocytes, and alkaline phosphatase—yielded negative results, contrary to our expectations. We have not yet found a reasonable explanation for this outcome, suggesting the need for further exploration of the potential mechanisms of glyphosate. Additionally, we discovered an intriguing result: subgroup analysis suggests that the harm of glyphosate exposure on ASCVD is higher in non-smokers, non-hypertensive patients, and individuals with higher activity levels, which contradicts our initial expectations and previous findings. This result may be due to confounding factors. Given the relative scarcity of research on the human toxicity of glyphosate, especially large-scale, long-term studies, further extensive research is needed to investigate this phenomenon.

As one of the most common pesticides, studying the toxic effects of glyphosate exposure on the human body has become an unavoidable issue [5–7]. Although many scholars have begun researching the human toxicity of glyphosate, exploring various systems including the nervous system [30], circulatory system [26], and reproductive system [31], some have delved into the microscopic level to understand its toxic mechanisms [23]. However, research on the association between glyphosate exposure and CVD remains insufficient, with no unified conclusion reached. Therefore, the establishment of this study is highly meaningful. We discovered that glyphosate exposure is a risk factor for ASCVD and speculated on its potential mechanisms within a reasonable range, laying the groundwork for future research in this field. Admittedly, our study has certain limitations. Firstly, in model construction, we only adjusted for basic diseases such as hypertension, diabetes, and hyperlipidemia, as we could not obtain specific results of cardiac-related examinations from patients, such as echocardiography or coronary angiography. This prevented us from adjusting for cardiac function as a confounding factor, potentially introducing bias. Secondly, disease establishment was based on patient self-reports rather than objective examination results, possibly leading to recall bias. Thirdly, because this study was cross-sectional, we could not effectively establish a causal relationship between glyphosate exposure and ASCVD. Although previous studies led us to infer glyphosate as the cause and ASCVD as the effect, this result was not generated by our experiment. Lastly, there are some results from our study that we currently cannot explain, necessitating further research on a larger scale, for a longer duration, and across multiple centers.

## Conclusion

In this retrospective cross-sectional study, we analyzed the association between glyphosate exposure and cardiovascular diseases. We found that as glyphosate exposure levels increased,

the risk of ASCVD steadily rose. Additionally, we discovered that BMI plays a mediating role in the association between glyphosate exposure and ASCVD. Furthermore, in subsequent regression analysis, we also identified glyphosate exposure as a risk factor for coronary heart disease, angina, and heart attacks. Our study provides further analysis of the toxic effects of glyphosate on the circulatory system and yielded promising results, but further research is still needed to validate our findings.

## Supporting information

**S1 Table. Basic characteristics of glyphosate-exposed population in 2013–2014 (grouped by presence of CVD).** Q1, 0–25%; Q2, 25%-50%; Q3, 50%-75%; Q4, 75–100%.
(DOCX)

**S2 Table. Basic characteristics of glyphosate-exposed population in 2013–2014 (grouped by presence of ASCVD).** Q1, 0–25%; Q2, 25%-50%; Q3, 50%-75%; Q4, 75–100%.
(DOCX)

**S3 Table. Multivariable linear regression of inflammatory markers, BMI, and glyphosate exposure levels.**
(DOCX)

**S4 Table. Mediating effects of inflammatory factors and BMI on the association between glyphosate exposure and ASCVD prevalence.**
(DOCX)

**S5 Table. OR (95%CI) of ASCVD according to different level of glyphosate exposure after further adjustment of several biomarkers.** *Q1, 0–25%; Q2, 25%-50%; Q3, 50%-75%; Q4, 75–100%. Model 4: Model 3 + Aspartatetransaminase, Alaninetransaminase, γ-glutamyl transpeptadase. Model 7: Model 3 + Serum creatinine, Urine creatinine, Blood urea nitrogen, Glomerular filtration rate, Urinary albumin creatinine ratio, Uric acid. Model 6: Model 3 + removing the missing values, Q1: 0–25.04%, Q2: 25.04–50%, Q3: 50–75.04%, Q4: 75.04–100%.
(DOCX)

## Acknowledgments

We thank all the staff who worked on the NHANES database and all the participants in this study. I would also like to thank all the authors for their efforts in finalizing the final version of this article.

## Author Contributions

**Conceptualization:** Wendi Xu, Zhe Chen, Yuhong Jiang, Hongbo Zeng, Nan He, Ziyi Liu, Meirong Zhou.

**Data curation:** Wendi Xu, Zhe Chen.

**Formal analysis:** Wendi Xu, Zhe Chen.

**Investigation:** Zhe Chen.

**Methodology:** Wendi Xu, Zhe Chen, Yuhong Jiang, Hongbo Zeng, Nan He, Ziyi Liu, Meirong Zhou.

**Project administration:** Wendi Xu, Zhe Chen, Meirong Zhou.

**Resources:** Meirong Zhou.

**Software:** Ziyi Liu.

**Supervision:** Meirong Zhou.

**Validation:** Ziyi Liu.

**Visualization:** Wendi Xu, Ziyi Liu.

**Writing – original draft:** Wendi Xu, Zhe Chen, Yuhong Jiang.

**Writing – review & editing:** Hongbo Zeng, Nan He, Ziyi Liu, Meirong Zhou.

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
