## [Decision Letter · Decision Letter 0]

4 Dec 2024

PONE-D-24-50752BMI-Mediated Association between Glyphosate Exposure and Increased Risk of Atherosclerotic Heart Disease: A Large-Scale Cross-Sectional StudyPLOS ONE

Dear Dr. Meirong Zhou,

Thank you for submitting your manuscript to PLOS ONE. After careful consideration, we feel that it has merit but does not fully meet PLOS ONE’s publication criteria as it currently stands. Therefore, we invite you to submit a revised version of the manuscript that addresses the points raised during the review process.

We look forward to receiving your revised manuscript.

Kind regards,

Hean Teik Ong

Academic Editor

PLOS ONE

Journal Requirements:

4. We note that Figure Graphical Abstract in your submission contain copyrighted images. All PLOS content is published under the Creative Commons Attribution License (CC BY 4.0), which means that the manuscript, images, and Supporting Information files will be freely available online, and any third party is permitted to access, download, copy, distribute, and use these materials in any way, even commercially, with proper attribution. For more information, see our copyright guidelines: http://journals.plos.org/plosone/s/licenses-and-copyright.

a. You may seek permission from the original copyright holder of Figure Graphical Abstract to publish the content specifically under the CC BY 4.0 license. 

Additional Editor Comments:

Authors should make minor revisions, addressing the points raised by the administrative editor and reviewers:

As this is an observational study of clinical data, please ensure compliance with the STROBE checklist (http://www.strobe-statement.org);

For studies with routinely-collected data, please ensure compliance with the RECORD checklist (http://record-statement.org).

The limitations of observational studies have been acknowledged, including in the abstract

Please ensure there are no unsupported statements of causation

Please ensure that the analysis is not affected by confounding variables, a lack of generalizability, selective reporting, post hoc analyses, or data dredging.

Reviewers' comments:

Reviewer's Responses to Questions

**Comments to the Author**

1. Is the manuscript technically sound, and do the data support the conclusions?

Reviewer #1: Yes

Reviewer #2: Yes

2. Has the statistical analysis been performed appropriately and rigorously? 

Reviewer #1: Yes

Reviewer #2: I Don't Know

3. Have the authors made all data underlying the findings in their manuscript fully available?

Reviewer #1: Yes

Reviewer #2: Yes

4. Is the manuscript presented in an intelligible fashion and written in standard English?

Reviewer #1: Yes

Reviewer #2: Yes

5. Review Comments to the Author

Reviewer #1: 1) The study highlights a significant association between glyphosate exposure and ASCVD but does not differentiate between the effects of acute and chronic exposure. Could the authors clarify whether the findings suggest a more prominent role for long-term, chronic exposure versus short-term, acute exposure in the development of ASCVD?

2) While BMI is presented as a mediator in this association, have the authors considered other potential mediators, such as inflammatory markers or oxidative stress pathways? Exploring these could provide a more comprehensive understanding of the mechanisms underlying the association between glyphosate exposure and ASCVD

3) Consider providing a more extensive comparison with global studies, particularly those investigating glyphosate exposure in high-use agricultural regions (e.g., Latin America, Southeast Asia) and among populations with varying baseline risks for ASCVD. This would enhance the contextualization of the study findings and highlight the broader relevance of the research.

Reviewer #2: Line 130-134 alcohol consumption classification, it would be good to use standardized NIAAA classification https://www.niaaa.nih.gov/alcohol-health/overview-alcohol-consumption/moderate-binge-drinking so that it is easier for comparison with other studies.

6. PLOS authors have the option to publish the peer review history of their article (what does this mean?). If published, this will include your full peer review and any attached files.

Reviewer #1: **Yes: **Sze Loon CHOW

Reviewer #2: No

---

## [Author Response · Author response to Decision Letter 0]

16 Dec 2024

Dear editors and reviewers,

Thank you so much for taking the time to review our article amidst your busy schedule and for providing valuable suggestions regarding the shortcomings in our research. Your insights have helped us recognize our deficiencies and identify areas for improvement. We greatly value your feedback and feel truly honored to have the opportunity for your review. We have carefully analyzed your comments, and below, you will find our responses to your review: 

Journal Requirements: 

The following are the revisions I have made in response to the journal requirements.

1.Thank you very much for pointing out the formatting issues in our submission. We sincerely apologize for not adhering strictly to the journal's style requirements in our initial submission. Following your guidance, we have revised the manuscript according to the PLOS ONE style template and carefully reviewed the updated version to ensure it complies with the journal's standards. At this point, we have not identified any other formatting errors in the revised document.

2.Regarding the issue with the corresponding author’s ORCID iD not being verified in Editorial Manager, I have now successfully completed the verification and updated the information in the system. My ORCID iD is currently displayed correctly in Editorial Manager.

3.Thank you very much for your reminder. In the revised version, I have relocated the ethical statement entirely to the Methods section of the manuscript and removed it from other parts of the text. The revised ethical statement is now located in lines 82–86 of the manuscript, as follows: 

“This research received approval from the Institutional Review Board of the National Center for Health Statistics, and all participants provided written informed consent. The study was conducted in accordance with the Declaration of Helsinki, and study protocol was approved by the NCHS Institutional Review Board (Continuation of Protocol #2011-17).”

4.Regarding the copyright concerns related to the graphical abstract, we have communicated with the copyright holder and obtained the necessary permissions. However, our team has revisited the relevance of the graphical abstract and held a meeting to discuss whether it is essential to the article and whether it serves its intended guiding purpose. The majority concluded that the graphical abstract does not seem to fulfill its intended role. As a result, we have decided to remove the previously submitted graphical abstract from the article. We hope you can understand our decision, and if you believe that removing the graphical abstract is inappropriate, please feel free to reach out to us for further discussion. 

5.We sincerely apologize for not noticing the submission requirements in our initial submission, which resulted in the omission of the supporting information at the end of the manuscript. Upon receiving your reminder, we have made the necessary corrections and have added the supporting information to the end of the manuscript, located on lines 460–478. Given the length of the original text, we have not included it in this response.

Additional Editor Comments: 

We greatly appreciate your support and recognition of our research; it is truly an honor for us. In response to the insights and concerns you have raised, we would like to provide the following clarifications:

1.We have thoroughly re-evaluated our manuscript against both the STROBE and RECORD checklists and can confirm that the manuscript fully adheres to their respective requirements.

2.We assure you that any causal claims within the manuscript are grounded in the study's results and are not speculative or unfounded.

3.During the analysis, we made every effort to account for and eliminate the influence of potential confounding factors, ensuring that the findings remain robust and unaffected by such variables.

4.We further guarantee that our results are generalizable, free from selective reporting, and supported by rigorous post-hoc analyses.

Once again, we deeply appreciate your recognition of our work. Should you have any further questions or additional valuable suggestions, please do not hesitate to contact us. 

Reviewer #1: 

1.We sincerely appreciate your willingness to review our manuscript and provide us with invaluable feedback. The question of whether long-term, chronic exposure or short-term, acute exposure plays a more prominent role in the development of ASCVD is indeed a thought-provoking and meaningful area of inquiry. Understanding this issue could further elucidate the mechanisms through which glyphosate influences the occurrence and progression of ASCVD. However, addressing this requires additional studies for validation. Our current study is a cross-sectional analysis, which allows us to explore the association between glyphosate exposure levels and ASCVD prevalence. Unfortunately, it does not enable us to differentiate between the effects of acute versus chronic exposure. Investigating this matter will be a key focus of our future research efforts. 

2.We believe that the issue of other potential mediators you raised is highly insightful and meaningful. During the course of our study, we also considered whether inflammatory factors might similarly act as mediators. To address this, we conducted an analysis to determine whether inflammatory factors mediated the association between glyphosate exposure and ASCVD prevalence. We selected three inflammatory markers for analysis: white blood cells, neutrophils, and lymphocytes. However, none of these markers demonstrated any significant mediating effects.

3.Thank you once again for your valuable suggestions. We deeply appreciate the originality and research significance of your input. The study participants in our research were limited to North America (United States), and regions with higher pesticide usage, such as Latin America and Southeast Asia, were not included. This is one of the limitations of our study. In response to your suggestion to broaden the scope by comparing our findings with global research, we conducted a new search using "glyphosate" as the key term. Unfortunately, we found no large-scale studies examining the relationship between glyphosate and ASCVD on a global scale. Additionally, global studies on glyphosate remain limited. Similar to the first suggestion you proposed, this represents a key direction for our future efforts. Moving forward, larger-scale, longer-term, and multi-center prospective studies will be essential.

Reviewer #2: 

We sincerely appreciate you taking the time out of your busy schedule to review our article and provide us with your invaluable feedback. Your insights are highly valued, as we believe they contribute significantly to controlling confounding factors, enabling meaningful comparisons with other studies, and enhancing the validity and applicability of our article. After consulting the standardized NIAAA classification, we noted that it categorizes alcohol consumption into six distinct types: 

Drinking in Moderation: Defined as no alcohol consumption or a daily intake of up to 2 standard drinks for men and 1 standard drink for women.

Binge Drinking: Characterized by a drinking pattern that brings blood alcohol concentration to 0.08% or higher within 2 hours, typically involving 5 or more drinks for men or 4 or more drinks for women on a single occasion.

High-Intensity Drinking: Refers to the consumption of 10 or more standard drinks for men or 8 or more for women during a single drinking occasion.

Heavy Drinking: Defined as consuming 5 or more drinks on any day or 15 or more drinks per week for men, and 4 or more drinks on any day or 8 or more drinks per week for women.

Alcohol Misuse: Encompasses unhealthy or harmful drinking behaviors that exceed recommended limits and result in physiological, psychological, or social consequences.

Alcohol Use Disorder: A medical diagnosis marked by impaired ability to stop or control alcohol consumption despite negative social, occupational, or health consequences.。

After thorough discussion within our team, we concluded that while the NIAAA classification has broad applicability, it does not appear to align with the specific needs of our research. The data for our study were obtained from NHANES, and this classification method does not seem to be applicable to this database. Furthermore, our categorization of alcohol-related covariates was informed by other relevant studies, one of which is referenced in this response for your convenience (DOI: 10.1002/iid3.70050).

Once again, we thank you for your valuable comments. Your feedback plays a crucial role in guiding us toward refining the categorization of covariates in our future research.

We sincerely appreciate all the valuable comments and recognition from the editors and reviewers, which have greatly supported and guided our research efforts. If you have any further questions, we would be delighted to receive your guidance again.

With kind regards,

Meirong Zhou

---

## [Decision Letter · Decision Letter 1]

7 Jan 2025

BMI-Mediated Association between Glyphosate Exposure and Increased Risk of Atherosclerotic Heart Disease: A Large-Scale Cross-Sectional Study

PONE-D-24-50752R1

Dear Dr. Meirong Zhou,

We’re pleased to inform you that your manuscript has been judged scientifically suitable for publication and will be formally accepted for publication once it meets all outstanding technical requirements.

Kind regards,

Hean Teik Ong

Academic Editor

PLOS ONE

Additional Editor Comments (optional):

Reviewers' comments:

Reviewer's Responses to Questions

**Comments to the Author**

1. If the authors have adequately addressed your comments raised in a previous round of review and you feel that this manuscript is now acceptable for publication, you may indicate that here to bypass the “Comments to the Author” section, enter your conflict of interest statement in the “Confidential to Editor” section, and submit your "Accept" recommendation.

Reviewer #1: All comments have been addressed

Reviewer #2: All comments have been addressed

2. Is the manuscript technically sound, and do the data support the conclusions?

Reviewer #1: Yes

Reviewer #2: Yes

3. Has the statistical analysis been performed appropriately and rigorously? 

Reviewer #1: Yes

Reviewer #2: I Don't Know

4. Have the authors made all data underlying the findings in their manuscript fully available?

Reviewer #1: Yes

Reviewer #2: Yes

5. Is the manuscript presented in an intelligible fashion and written in standard English?

Reviewer #1: Yes

Reviewer #2: Yes

6. Review Comments to the Author

Reviewer #1: Authors have addressed and answered the reviewer's comments. The research paper highlighted the importance of BMI as a mediator in developing Atherosclerotic Heart Disease

Reviewer #2: Your research paper on BMI-Mediated Association between Glyphosate Exposure and Increased Risk of

Atherosclerotic Heart Disease is a relevant since CVD is the top cause of mortality in the world, and obesity is increasingly pervasive in all society.

7. PLOS authors have the option to publish the peer review history of their article (what does this mean?). If published, this will include your full peer review and any attached files.

Reviewer #1: **Yes: **CHOW Sze Loon

Reviewer #2: No

---

## [Editor Report · Acceptance letter]

14 Jan 2025

PONE-D-24-50752R1 

PLOS ONE

Dear Dr. Zhou, 

I'm pleased to inform you that your manuscript has been deemed suitable for publication in PLOS ONE. Congratulations! Your manuscript is now being handed over to our production team.

Kind regards, 

on behalf of

Dr. Hean Teik Ong 

Academic Editor

PLOS ONE